# The Chloronium Cation [(C_2_H_3_)_2_Cl^+^] and Unsaturated C_4_-Carbocations with C=C and C≡C Bonds in Their Solid Salts and in Solutions: An H^1^/C^13^ NMR and Infrared Spectroscopic Study

**DOI:** 10.3390/ijms23169111

**Published:** 2022-08-14

**Authors:** Evgenii S. Stoyanov, Irina V. Stoyanova

**Affiliations:** Vorozhtsov Institute of Organic Chemistry, Siberian Branch of Russian Academy of Sciences, 630090 Novosibirsk, Russia

**Keywords:** chloronium cation, vinyl and propargyl carbocations, magic-angle spinning NMR measurements

## Abstract

Solid salts of the divinyl chloronium (C_2_H_3_)_2_Cl^+^ cation (**I**) and unsaturated C_4_H_6_Cl^+^ and C_4_H_7_^+^ carbocations with the highly stable CHB_11_Hal_11_^−^ anion (Hal=F, Cl) were obtained for the first time. At 120 °C, the salt of the chloronium cation decomposes, yielding a salt of the C_4_H_5_^+^ cation. This thermally stable (up to 200 °C) carbocation is methyl propargyl, CH≡C-C^+^-H-CH_3_ (**VI**), which, according to quantum chemical calculations, should be energetically much less favorable than other isomers of the C_4_H_7_^+^ cations. Cation **VI** readily attaches HCl to the formal triple C≡C bond to form the CHCl=CH-C^+^H-CH_3_ cation (**VII**). In infrared spectra of cations **I**, **VI**, and **VII**, frequencies of C=C and C≡C stretches are significantly lower than those predicted by calculations (by 400–500 cm^−1^). Infrared and ^1^H/^13^C magic-angle spinning NMR spectra of solid salts of cations **I** and **VI** and high-resolution ^1^H/^13^C NMR spectra of **VII** in solution in SO_2_ClF were interpreted. On the basis of the spectroscopic data, the charge and electron density distribution in the cations are discussed.

## 1. Introduction

Simple saturated carbocations (C_2_–C_7_) have been studied experimentally in condensed phases [1,2,3,4,5,6,7] and their infrared (IR) spectra have been interpreted [6,7,8]. Protonated arenium cations, starting with C_6_H_7_^+^, have been broadly studied by NMR spectroscopy in liquid superacids and as carborane salts [9,10]. There are more problems with the research on unsaturated nonarenium carbocations, because their salts have not been obtained so far. The simplest and least stable vinyl cation, C_2_H_3_^+^, and isomers of C_3_H_3_^+^ and C_3_H_5_^+^ have been studied experimentally only in vacuum by mass-selected IR spectroscopy [11,12,13]. Their stable isomers have been identified and IR spectra interpreted. Numerous attempts have been made to study the allyl cation, C_3_H_5_^+^, by NMR spectroscopy in liquid superacids at low temperature, and they have failed [14]. The formation of C_3_H_5_^+^ has been proved in a cryogenic superacidic matrix (170 K) by IR spectroscopy [15,16]. With the increasing temperature (230 K), the IR spectrum changes and this phenomenon is attributed to “polymerization”. The main disadvantages of the conventional use of superacids such as FSO_3_H/SbF_5_ are the high oxidation potential and reactivity of the Lewis acid used, SbF_5_, which lead in the case of vinyl cations to rearrangement or decomposition products at temperatures above −100 °C [17,18]. Nevertheless, NMR spectroscopy has revealed the formation (in liquid superacids) of unstabilized C_5_-dienyl cations with two conjugated C=C bonds at −135 °C [18] and more stable cyclobutenyl and dimethyl-allyl cations [19,20,21]. The vinyl cations stabilized by β-silyl and other electron-donating groups were studied in the 1990s by H.-U. Siehl and coauthors [22,23,24] by NMR spectroscopy in liquid superacids at temperatures below −100 °C and were isolated as carborane salts [25]. Thus, from refs. [15,16,17,18,19,20,21,22,23,24,25], it follows that nonstabilized vinyl cations are stable only at low temperatures (less than −100 °C), and their salts are not isolable in a pure state. Recently, it was shown that this is not the case: carborane salts of the vinyl and allyl types of carbocations C_3_H_5_^+^ and C_4_H_7_^+^, stable at room and elevated temperatures (up to 140–150 °C), were obtained and characterized by X-ray diffraction analysis and IR spectroscopy [26,27,28]. The possibility of the existence of carbocations containing C≡C bonds is discussed on the basis of quantum chemical calculations [29], but they have not been obtained and characterized experimentally.

In this work, we report obtaining pure solid salts—stable at room temperature—of the unsaturated carbocations with double C=C (C_4_H_6_Cl^+^) and triple C≡C bonds (C_4_H_5_^+^) and a so far unknown divinyl chloronium cation, (C_2_H_3_)_2_Cl^+^. As counterions, carborane anions CHB_11_Hal_11_^−^ were chosen, where Hal=F, Cl (hereafter abbreviated as {F_11_^−^} and {Cl_11_^−^}, Figure 1) because of their extreme stability and minimal basicity, which promotes the formation of stable salts with highly reactive cations [30]. The cations being studied were characterized by ^1^H/^13^C NMR and IR spectroscopy.

## 2. Results and Discussion

### 2.1. The Chloronium Cation

It is formed by a three-stage reaction (see the Experimental Section). The first two stages
H {Cl_11_} + C_2_H_4_Cl_2_ → C_2_H_4_Cl^+^ {Cl_11_^−^} + HCl
and
C_2_H_4_Cl^+^ {Cl_11_^−^} + C_2_H_4_Cl_2_ → (C_2_H_4_Cl-Cl^+^-C_2_H_4_Cl) {Cl_11_^−^}
are similar to those studied for reactions of H{Cl_11_} with CH_2_Cl_2_ [31] or C_2_H_5_Cl [32] and should lead to the formation of the salt of the dichloroethylchloronium cation, C_2_H_4_Cl-Cl^+^-C_2_H_4_Cl. It decomposes rapidly with the release of two HCl molecules to form mostly expected divinylchloronium (third stage):(C_2_H_4_Cl-Cl^+^-C_2_H_4_Cl) {Cl_11_^−^} → (C_2_H_3_-Cl^+^-C_2_H_3_) {Cl_11_^−^} + 2HCl.

It will be shown below that in the IR spectra of the obtained salt, characteristic bands of chloronium group C-Cl^+^-C are observed at 628 and 594 cm^−1^, thereby unambiguously proving the formation of the chloronium cation.

The optimized structure of the cis-isomer of divinylchloronium at the B3LYP/6-311G++(d,p) level of theory is shown in Figure 1 (isomer **I**) with some geometrical parameters (the *trans*-isomer is almost the same in terms of energy). The presence of two double C=C bonds in cation **I** does not rule out that under certain conditions, intramolecular cyclization with the formation of isomer **II** may occur (Figure 1). The optimization of structure **II** shows that the C····C distance in the chloronium C–Cl^+^–C group decreases so much that the C–C bond and the four-membered carbon cycle can form (isomer **III**).

Energetically, cations **II** and **III** are more favorable than **I** by 31.2 and 11.3 kcal/mol, respectively.

***NMR spectra.*** The magic-angle spinning (MAS) ^13^C NMR spectrum of the isotope-substituted ^13^C-(C_2_H_3_)_2_Cl^+^{Cl_11_^−^} salt (99 atom% ^13^C) with high-power ^1^H decoupling contains four signals (Figure 2). Although the integration of ^13^C NMR signals is not recommended, we will point out that in this spectrum, the intensities of the signals at (83.2 + 78.6), 60.6 and 43.9 ppm correlate as 2.0:0.95:1.0, respectively. The MAS NMR ^13^C spectrum registered without ^1^H decoupling shows that two signals at 83.2 and 78.6 ppm are split with a ^1^*J*_CH_ constant of 304 Hz (Figure 2, red), which indicates that they belong to the C atoms of nonequivalent C_α_H groups (Figure 1). The other two signals obviously belong to carbon atoms of nonequivalent C_β_H_2_ groups. They must have a triplet structure. Nevertheless, due to the weaker influence of the positive charge on them, the spin–spin ^13^C–^1^H constant decreases, thereby leading to signal broadening without multiplet resolution. Thus, the chloronium cation contains two nonequivalent CH_2_CH groups.

The ^1^H MAS NMR spectrum of the (C_2_H_3_)_2_Cl^+^{Cl_11_^−^} salt (with the natural abundance of the ^13^C isotope) shows the known signal from the CH group of the {Cl_11_^−^} anion at 3.37 ppm and overlapped signals from the cation (Figure 3). In the ^1^H MAS NMR spectrum of the isotope-substituted ^13^C-(C_2_H_3_)_2_Cl^+^ {Cl_11_^−^} salt, signals from the cation are broadened due to spin–spin ^1^H–^13^C coupling without fine structure resolution. To obtain a high-resolution spectrum, the ^12^C-(C_2_H_3_)_2_Cl^+^{Cl_11_^−^} salt was dissolved in SO_2_ClF and a ^1^H NMR spectrum was registered. It turned out to be-time dependent: the HCl signal appears at 1.06 ppm and its intensity increases until the intensity of the CH signal of the {Cl_11_^−^} anion is reached. This means that chloronium in solution decomposes with the release of one HCl molecule:(C_2_H_3_)_2_Cl^+^{Cl_11_^−^} → C_4_H_5_^+^{Cl_11_^−^} + HCl.

The ^1^H NMR spectrum of the freshly prepared solution (recorded within 2–3 min after solution preparation at room temperature) shows strong signals from the chloronium cation and weak signals from HCl and the products of decomposition (Figure 4). Because a solid suspension in a liquid does not have time to completely precipitate within 2–3 min, the resolution of the spectrum is low and the ^3^*J*_HH_ decoupling is not observed. The spectrum shows all six signals expected for that of the asymmetric cation (C_2_H_3_)_2_Cl^+^. They correlate with the signals in the ^1^H MAS NMR spectrum subjected to separation into six Lorentzian components (Figure 3). Therefore, the ^1^H NMR spectra of chloronium are similar for a solution and the solid salt, and cation structure is the same in both phases.

The finding that each C_2_H_3_ group of the (C_2_H_3_)_2_Cl^+^ cation yields three H^1^ signals (H*_a_*, H*_b_*, and H*_c_*) that are known for neutral vinyl chloride with hindered rotation of the CH_2_ group around the C=C bond [33] means that C_2_H_3_ groups contain a double bond. That is, the cation under study is distorted divinylchloronium with two nonequivalent C_2_H_3_ groups. Its asymmetry in solid salt (C_2_H_3_)_2_Cl^+^{Cl_11_^−^} and in solutions may be due to the emergence of contact ion pairs with cation–anion interaction via the C_α_H group (Figure 2). In neutral vinyl chloride, the C_α_ atom of CH_2_C_α_HCl is mostly deshielded because it is mainly affected by the electronegativity of the Cl atom, while atoms of the CH_2_ group are deshielded less [33,34] (Table 1). The same is observed in cation **I**: the C_α_ atom closest to Cl^+^ is more deshielded than the C_β_ atom. There is also an important difference: the screening of H and C atoms in the chloronium cation is greater than that of vinyl chloride, i.e., the charge of the cation contributes to an increase in the screening constant of the C and H atoms.

***IR spectra.*** In Figure 5, IR spectra of the salts of protio and deutero chloronium cation are given. To identify the frequencies of CH, CC, and CCl vibrations, the difference in the spectra between the protio and deutero samples was obtained with such a scaling factor as to fully compensate for the adsorption of the {Cl_11_^−^} anion. This difference allowed us to determine the bands of the protio cation with positive intensities and those of the deutero cation as negative in the entire frequency range. The frequencies involving the C–H (D) vibrations yielded a common H/D isotopic ratio of 1.326 to 1.367 (Table 2). For the split band at 1238 cm^−1^, this ratio is 1.024, confirming its assignment to the CC stretch. Two C-Cl stretches of the bridged C-Cl^+^-C group at 628 and 594 cm^−1^ (very specific for chloronium cations [31,32]) remained nearly the same, as did another one at 740 cm^−1^, confirming that it did not involve vibrations from the H atom (can be a ClCC bending vibration).

The IR spectrum of the chloronium cation has two specific features. The first and most important is the absence of a strong C=C stretching band expected at ~1600 cm^−1^. Instead, it shows a split band of medium intensity at 1238 and 1234 cm^−1^, which is even lower than that of the t-Bu^+^ cation (~1290 cm^−1^). If this is indeed the band of the CC stretch vibration of cation **I**, then it is slightly stronger than the strengthened single C-C bond in the t-Bu^+^ cation, but weaker than the aromatic CC bond with one-and-half-bond status.

The second feature is the multiplet structure of the bands of CH_2_/CD_2_ stretch vibrations (Figure 5, Table 2); this arrangement is caused by the nonequivalence of the two C_2_H_3_ groups of cation **I** owing to ion pairing. In the salts with carborane anion {F_11_^−^}, which is much less basic than {Cl_11_^−^}, the ion pairing should weaken or not form at all. The IR spectrum of the C_4_H_6_Cl^+^{F_11_^−^} salt contains three CH stretch vibrations of the cation (Figure 6). Their frequencies are strictly proportional to the frequencies of vinyl chloride [35]: ν_as_CH_2_ 3121 cm^−1^, ν_s_CH_2_ 3030 cm^−1^ and νCH 3086 cm^−1^. These data allow their assignment (Figure 6) corresponding to symmetrical cation **I**. That is, the chloronium cation is uniformly surrounded by {F_11_^−^} anions. In the region of expected C=C stretching frequencies, a very weak band at 1607 cm^−1^ is observed. Nevertheless, when the salt is heated to 100 °C and chloronium is completely decomposed with the release of HCl (see Experimental section), a new intense νC=C band of the decomposition product emerges at 1567 cm^−1^, and a weak band at 1607 cm^−1^ remains unchanged (Figure 6). This result indicates that the latter band belongs to one of the formed by-products. Thus, the symmetric chloronium cation, just as the asymmetric one, does not show the absorption of the stretching vibration of the C=C double bond in the expected frequency range.

A comparison of the experimental spectrum of chloronium with those calculated for optimized structures of isomers **I**–**III** at the B3LYP/6-311G++(d,p) level of theory (Appendix A) does not show good agreement (especially for C=C stretch frequencies), which is generally typical for unsaturated carbocations [26,27].

Nevertheless, it is useful to compare the experimental stretch vibrations of the C–Cl^+^–C group, which are the most specific indicators of chloronium cations, with those calculated for **I**–**III** isomers, given that the ν_calc_/ν_exp_ ratio must be close to 1.0. For isomer **I**, empirical frequencies are closest to calculated ones with the ν_calc_/ν_exp_ ratio of 0.91 (Table 3). For isomers **II** and **III**, this ratio is unsatisfactorily low, which is a sufficient reason to exclude them from consideration.

### 2.2. The C_4_-Carbocations

Salts of the chloronium cation decompose at temperatures above 100 °C with the release of HCl and the formation of carbocation C_4_H_5_^+^ (see the Experimental part). Three noncyclic isomers—**IV**, **V**, and **VI**—with the lowest energy are possible (the energy of **V** and **VI** exceeds that of **IV** by 5.8 and 8.00 kcal/mol, respectively) [29]. Their optimized structures at the B3LYP/6-311G++(d,p) level of theory are depicted in Figure 3 with CC distances. Let us consider with which isomer the NMR and IR spectra of the C_4_H_5_^+^ cation are the most consistent.

***NMR spectra.*** The ^1^H MAS NMR spectrum of the isotope-substituted ^13^C-C_4_H_5_^+^{Cl_11_^−^} salt is given in Figure 7. It features a signal from the (^12^C)-H atom of the anion at 3.37 ppm and three signals from the cation. The intensity of the strongest signal at 2.55 ppm exceeds that of the anion by 3.3-fold, indicating that it belongs to the CH_3_ group of the cation. Two other signals of equal intensity at 7.44 and 9.35 ppm obviously belong to two CH groups of the cation. This result clearly supports the formation of isomer **VI**. The signal at 7.44 ppm is a doublet with a ^1^*J*_CH_ coupling constant of 184 Hz. Other cation signals should also be unresolved doublets because all H atoms are bonded to one ^13^C atom.

The ^1^H NMR spectrum was separated into Lorentzian components, taking into account the fact that the cation signals are doublets, and the anion signal is a singlet (Figure 7). A well-resolved doublet at 7.44 ppm may belong to the H atom of the H-(C≡C) group, and the signal at 9.35 ppm should be attributed to the H atom of the CH[CH_3_] group, because the broadening of the doublet components due to the spin–spin interaction with hydrogen atoms of the methyl group impairs their resolution.

^13^C MAS NMR analysis with high-power ^1^H decoupling yields four signals of this sample (Figure 8). The ^13^C NMR spectrum registered without ^1^H decoupling does not result in the manifestation of a fine structure owing to ^13^C-^1^H spin–spin interactions. The signal with a chemical shift of 36 ppm is typical for the CH_3_ group in carbocations. To interpret the remaining signals, it is desirable to see their hyperfine structure. For this, it is necessary to record the NMR spectra of salt solutions, if possible.

The salt of cation **VI** was dissolved in SO_2_ClF. Gaseous SO_2_ClF was liquefied in an NMR tube at −15 °C using a stream of gaseous SO_2_ClF passed through a desiccant P_2_O_5_. The ^1^H NMR spectrum of the solvent showed a weak broad signal at 8.90 ppm from H^+^(H_2_O)_n_, and a second one at 1.06 ppm from molecular HCl, which is formed upon the decomposition of SO_2_ClF with water. That is, the solvent contained HCl and H^+^(H_2_O)_n_Cl^−^ as impurities. The solution was obtained by the anaerobic condensation of SO_2_ClF in an NMR tube with the ^13^C-C_4_H_5_^+^{Cl_11_^−^} salt at the bottom; the tube was sealed and shaken until a saturated solution was obtained. Its ^1^H NMR spectrum contained weak solute signals along with a weak H^+^(H_2_O)_n_ signal that broadened and shifted to 7.1 ppm, indicating a decrease in water protonation. At the same time, the HCl signal almost disappeared. All of these data mean that HCl joined the cation.

The ^1^H NMR spectrum of the solute shows a known singlet from the CH group of the anion (3.37 ppm) and four signals from the cation, which are split into doublets due to the ^1^H−^13^C spin–spin interaction (Figure 9). The integrals of the cation signals strictly conform to the 1:1:1:3 ratio. Thus, this carbocation is not the same as that in the original solid salt. It has one CH_3_ group (at 2.69 ppm) and three CH groups. Obviously, upon salt dissolution, the HCl molecule joins the formal triple CC bond of cation **VI** to form the C_4_H_6_Cl^+^ cation **VII** (Figure 4).

In the ^1^H NMR spectrum of cation **VII**, the signal of the CH_3_ group at 2.69 ppm is a doublet with ^1^*J*_H-C_ = 130 Hz. Each doublet component is split into a quartet because of a spin–spin interaction of ^1^H with nucleus H_α_, C_α_ and C_β_ with ^2^*J*_HCα_ ≈ ^3^*J*_HHα_ ≈ ^3^*J*_HCβ_ ≈ 7 Hz. Two other signals at 7.04 ppm and 8.42 ppm with constants ^1^*J*_H-C_ = 168 and 158 Hz, respectively, are characteristic of the CH groups with sp^2^ hybridization of the carbon atom and belong to the middle groups, C_α_H and C_β_H. The doublet at 8.42 ppm has a quintet-like structure due to the spin–spin interaction of the H_α_ atom with the protons of the CH_3_ and CH_β_ groups and the C(H_3_), C_β_ and C_γ_ carbon atoms. If all of the constants of these interactions are approximately the same and equal to 7 Hz, then the splitting will be quintet. The doublet signal at 7.04 ppm with ^1^*J*_H-C_ = 168 Hz belongs to the H_β_ atom. Its multiplicity is poorly resolved due to the spin–spin interaction of the H_β_ atom with more differentiated atoms of groups C_α_H and C_γ_H. Finally, the signal at 9.58 ppm with ^1^*J*_CH_ 183 Hz does not have ultrafine structure and belongs to the terminal C_γ_H(Cl) group. Its increased ^1^*J*_C__γH_ compared to ^1^*J*_C__βH_ (168 Hz) can be explained by the high electronegativity of the Cl atom bound to C_γ_.

The ^13^C NMR spectrum of the same solution recorded with broad-band decoupling ^13^C {^1^H} shows four signals (Figure 10, black). The signal at 20.8 ppm is a doublet with ^1^*J*_CC_ = 39 Hz. It belongs to the sp^3^ C atom of the methyl group interacting with the C_α_ atom’s spin with sp^2^ hybridization. The components of this doublet are also split into doublets owing to a long-range interaction with C_β_, with ^2^*J*_CC__β_ = 7 Hz. The signal at 129 ppm is a triplet belonging to the C_β_ atom bound to C_α_ and Cγ atoms with ^1^*J*_C-C_ = 60 Hz, typical of the *J*_Csp__2_-_Csp__2_ constants. The signal at 186 ppm is a superposition of two doublets with ^1^*J*_C-C_ = 60 Hz. It belongs to the C_α_ atom bound to significantly different atoms, C(H_3_) and C_β_. Finally, the doublet signal at 203 ppm with the constant ^1^*J*_C-C_ = 60 Hz belongs to the C_γ_ atom of the C_γ_HCl terminal group.

The monoresonance ^13^C NMR spectrum of this solution is in agreement with the signal assignment proposed above (Figure 10, blue). The signal of the CH_3_ group is split into a triplet with ^1^*J*_C-H_ = 130 Hz (^1^*J*_C-C_ = 39 Hz); the C_β_ signal is split into two triplets with ^1^*J*_C-H_ = 168 Hz (^1^*J*_C-C_ = 60 Hz); two doublet C_α_ signals are doubled with ^1^*J*_C-H_ = 158 Hz (^1^*J*_C-C_ = 60 and 39 Hz) and the signal from the C_γ_ atom shows a split doublet with ^1^*J*_C-H_ = 183 Hz and ^1^*J*_C-C_ = 60 Hz. All data from the ^13^C NMR spectrum of cation **VII** are summarized in Table 4.

To test whether there was a similarity between the chemical shifts of the ^13^C signals of cations **VII** and **VI**, the interpreted signals from **VII** were compared with those from **VI**. An unexpectedly good correlation was obtained (Figure 11), which implies that the shielding of ^13^C nuclei in carbon skeletons 
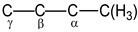
 of both cations was similar. Based on this similarity, it is possible to interpret the ^13^C signals of cation **VI** (Table 5).

***IR spectra.*** The formation of the C_4_H_5_^+^ cation from chloronium at 120 °C results in the disappearance of the bands of the most characteristic stretch vibrations of the C-Cl^+^-C moiety of chloronium, and in the appearance of a new spectrum with a highly intense band at 1563 cm^−1^, typical of the stretching vibrations of the multiple CC bond of unsaturated hydrocarbons (Figure 12). The sequential recording of IR spectra with increasing temperature does not yield absorption bands of any intermediates, except for the spectrum of the final product, which increases in intensity. This means that if intermediate products are formed, they are unstable and quickly transform into the final product. It is expected that the decomposition of chloronium produces isomer **IV**, which, according to quantum chemical calculations, has the lowest energy among all the open-chain isomers, **IV**–**VI [29]**. Nevertheless, according to NMR spectra, cation **VI** is formed. Therefore, if isomer **IV** comes into being at the first stage, then it quickly transitions into isomer **VI** (Figure 5).

Stronger heating of the C_4_H_5_^+^{Cl_11_^−^} salt to 200 °C does not induce any changes in the IR spectrum of carbocation **VI**, indicating its thermal stability. The IR spectra of C_4_H_5_^+^ are similar between its salts with {F_11_^−^} and {Cl_11_^−^} anions (Appendix A), which means that its structure is not affected by any change in the basicity of the anionic environment. Because the absorption pattern of the {F_11_^−^} anion overlaps with a larger part of the C_4_H_5_^+^ spectrum, as compared to that of {Cl_11_^−^}, we discuss the spectrum of C_4_H_5_^+^{Cl_11_^−^} salt.

In Figure 12, the IR spectra of salts C_4_H_5_^+^{Cl_11_^−^} and C_4_D_5_^+^{Cl_11_^−^} are depicted. Their difference, obtained via the complete compensation of the absorption of the {Cl_11_^−^} anion, allows detecting all bands of the protio-cation with positive intensity and those of the deutero-cation with negative intensity. Pairs of bands with the H/D isotopic ratio of 1.34–1.36 belong to CH vibrations (Table 6). For the intense band at 1563 cm^−1^ and the middle one at 1447 cm^−1^ (protio-sample), this ratio is ~1.036 and 1.057, respectively, which means that they belong to the stretching vibrations of CC bonds. Their frequencies approach the double and one-and-a-half bond status, respectively, and the formal structure of the cation **VI** is more correctly represented as shown in Figure 6. On the other hand, the CC stretch frequencies of cation **VI** can be considered as asymmetric (1563 cm^−1^) and symmetric (1447 cm^−1^) CCC vibrations. Then, structure **VI** can be described as resonance between two structures **VIa** and **VIb** (Figure 6).

The IR spectrum of unsaturated cation **VII** was found to have low intensity because its salt was obtained in a small amount by the evaporation of the solvent from its solution in SO_2_ClF. Nevertheless, the most important frequencies are detectable with high confidence (Appendix A). The IR spectrum of **VII** does not contain an intense band of C=C stretching above 1600 cm^−1^, which is present in the other studied isomers of the C_4_H_6_Cl^+^ cation, **VIII** (at 1680 cm^−1^) and **IX** (at 1710 cm^−1^), with π-electron density concentrated mainly on the C=C bond [28] (Figure 7). Instead, its IR spectrum shows two bands of CC stretches at 1440 and 1262 cm^−1^, whose low frequencies are more consistent with symmetrical and asymmetrical vibrations, respectively, of allyl moiety 
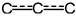
. The frequency of the C-Cl stretch vibration is seen at 636 cm^−1^ (Appendix A), which is much lower than that of isomers **VIII** (at 768 cm^−1^) and **IX** (at 733 cm^−1^) [28] and matches the frequency of asymmetric vibration of the bridged C–Cl^+^–C group of the stablest dimethylchloronium cation, which decomposes at elevated temperature [32]. Therefore, the C-Cl bond is weak, and with an increase in temperature, cation **VII** can lose the HCl molecule.

### 2.3. Comparative Discussion of the IR and NMR Spectroscopic Data

#### 2.3.1. The Chloronium Cation

Even though, according to calculations, cyclic cation **III** is energetically more favorable than open-chain **I** by 31.2 kcal/mol, it is **I** that forms in the solid salt and exists in solutions. A specific feature of cation **I** is its asymmetry in salts with {Cl_11_^−^} anions, owing to the ion pairing with the anion. In SO_2_ClF solutions, cation **I** retains its asymmetry due to the formation of contact ion pairs. In a salt with the least basic {F_11_^−^} anions, the IR spectrum of the cation corresponds to a symmetrical one, as predicted by quantum chemical calculations for a naked cation **I** in vacuum. Such a high sensitivity of the cation to the influence of the environment indicates its high polarizability.

The asymmetry of cation **I** in ion pairs arises due to its interaction with the anion, not through its Cl atom, but rather through the C_α_H group, as illustrated in Figure 2. In other words, the positive charge is more substantial on the C_α_H group.

The asymmetry of the cation is manifested mostly in the vibrations of its CH stretches rather than in those of its CC stretches, which differ only by 5 cm^−1^. A comparison of its IR spectrum with the interpreted spectrum of neutral vinyl chloride [35] (without C-Cl and CCCl vibrations) allows the assignment of its most characteristic frequencies (Appendix A)), and shows a good correlation for all frequencies with the exception of that for C=C stretching (Figure 13), which deviates downward from the correlation line by 340 cm^−1^. The same discrepancy is observed when we compare the experimental and calculated C=C stretching frequencies of cation **I**. The calculated νC=C of 1602 cm^−1^ (Appendix A after scaling by a factor of 0.9674) is even slightly greater than that of neutral vinyl chloride and exceeds the experimental one for **I** by 360 cm^−1^. Put another way, the positive charge of cation **I** has the strongest effect on its C=C bond, and this phenomenon is poorly described by quantum chemical calculations. The nature of the low frequency of C=C stretching vibrations and its high sensitivity to the presence of electron-donating substituents in the cation are discussed in the [28].

Cation **I** also possesses unusual features in NMR spectra: the impact of the positive charge on its C_2_H_3_- groups increases the screening constants of the C atoms while weakening the C=C bond, compared to those of neutral C_2_H_3_Cl. Thus, a positive charge reduces the π-electron density between carbon atoms, bringing its nature closer to a single σ-bond (the screening of C atoms increases).

A comparison of cation **I** with its saturated analog, diethylchloronium (C_2_H_5_)_2_Cl^+^ [32], suggests that frequencies of symmetric and asymmetric stretching vibrations of the chloronium C-Cl^+^-C group in **I** are higher by 80 and 83 cm^−1^, respectively. This is a large difference. It means that the C-Cl^+^-C group attracts electron density from the C_2_H_5_- group, thereby imparting a weak π-character to C-Cl bonds and reducing the π-character of the C=C bond. Thus, the C_2_H_3_ group under the influence of a positive charge becomes an effective electron donor, attracting a positive charge.

#### 2.3.2. The C_4_-Carbocations

When chloronium decomposes at >120 °C, the most stable isomer, **IV**, forms first (Figure 5). Nonetheless, if this happens, then **IV** quickly converts to **VI**, which, according to quantum chemical calculations at the B3LYP/6-311G++(d,p) or MP2(full)/6-311G(d,p) levels of theory, is energetically less favorable by 8.0 or 7.6 kcal/mol [29], respectively. The calculations also predict that the transition of **IV** to **VI** takes place in two stages, with an activation energy of 39 kcal/mol [29], which should prevent the formation of **VI**. Nevertheless, the formation of isomer **VI** was revealed to be the most preferable.

Isomer **VI** in carborane salts can be in a homogeneous and inhomogeneous (with predominant interaction with one counterion) anionic environment. Both states, differing in νC=C frequencies, have been found for vinyl-type carbocations [26,27,28]. On the other hand, isomer **VI** in salts with {Cl_11_^−^} and with the least basic {F_11_^−^} anion shows similar frequencies of C≡C stretching, pointing to its homogeneous anionic environment.

The IR spectrum of isomer **VI** has two features related to the vibration frequencies of C≡C and C-H stretches. The first one is νC≡C at 1563 cm^−1^, which is 73 cm^−1^ higher than that of the C=C double bond (1490 cm^−1^) of cations (CH_3_)_2_C=C^+^H or CH_3_C^+^=CH [26,27], and this is expected. Nonetheless, the calculated C≡C stretch (at B3LYP/6-311G++(d,p) level of theory, with a scaling of 0.967) is 2020 cm^−1^, which significantly exceeds the experimental one by 450 cm^−1^. A similar discrepancy between the calculated and experimental values of the C=C stretching frequencies equal to 362–246 cm^−1^ has been documented for vinyl carbocations, CH_3_-CH=CH^+^ and CH_3_-C^+^=CH_2_, respectively [26,27].

The second feature of the IR spectra of **VI** is the reduced frequency of stretching vibrations of its CH_3_ group by ~120 cm^−1^ relative to those of neutral alkanes. This is possible if the CH_3_ group is involved in hyperconjugation with the partially empty 2p_z_ orbital of the C_α_ atom. The decrease in frequency is not very large compared to that observed for *t*-butyl^+^ and *i*-propyl^+^ alkane carbocations (~160 cm^−1^), which have a well-pronounced hyperconjugation effect [6]. This observation clearly indicates that in isomer **VI**, the triple C≡C bond is partially delocalized (Figure 6) with the predominant contribution of resonance structure **VIa**, because a weak contribution of resonance structure **VIb** promotes the weak filling of the 2p_z_ orbital of the C_α_ atom and the hyperconjugation effect is partially attenuated.

The calculated IR spectrum of isomer **VI** revealed that its CH_3_ group should not be disturbed by hyperconjugation (Appendix A). Similarly, the calculated frequency of the C_γ_H group at 3137 cm^−1^ is typical for the H-C≡C group of neutral molecules and exceeds the empirical one (at 3058 cm^−1^) by ~80 cm^−1^. Therefore, the application of quantum chemical calculations to the interpretation of IR spectroscopic data should be carried out with great care. The same conclusion was drawn earlier in a study on vinyl-type carbocations [26,27].

The obtained NMR spectra allow us to discern some pattern in their changes for alkanes, alkenes, and unsaturated carbocations. A strongly deshielded ^13^C NMR signal (>150 ppm relative to TMS) can be confidently assigned to a positively charged carbon atom. The largest chemical shift has been documented for the central carbon atom of the *tert*-butyl cation at 335.2 ppm [36]. In unsaturated carbocations, the C=C double bond is often delocalized, forming allyl moiety 
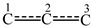
. Chemical shifts of its C atoms usually vary depending on the substituents attached to it: from 170 to 190 ppm for C_1,3_, and 170–150 ppm for C_2_ atoms, respectively [37], thus pointing to greater charge localization on atoms C_1_ and C_3_. If the double bond is not delocalized, as for example, in the 
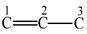
 moiety of the 1-cyclopropylcyclopropyli-dene methyl cation, then the positive charge is located mainly on the C_2_ atom with a chemical shift of 234 ppm and less on atoms C_1_ (51.7 ppm) and C_3_ (21.2 ppm) [38]. In vinyl-type carbocations R′C^+^=CR″_2_ stabilized by electron-donating groups R′ and R″, such as 
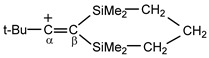
, the positive charge on the C=C bond is greatly reduced due to the combined influence of t-Bu and the two β-silyl substituents directly attached to it [39]. In the C^13^ NMR spectra, the signal from the C_α_ atom is observed in a weak field at 202.7 ppm and from the C_β_ atom at 75.5 ppm, that is, the charge is mainly localized on the C_α_ atom.

Vinyl cation **VII**, formally studied by us, has a double C=C bond in the 
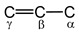
 moiety, whose carbon atoms possess the chemical shifts C_γ_ (203 ppm), C_β_ (129 ppm) and C_α_ (183 ppm). They do not match a triatomic moiety with one CC double bond but rather correspond to slightly asymmetric allyl moiety 
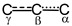
. Therefore, it is more correct to represent structure **VII** as shown in Figure 8.

Cation **VI** formally contains a triple CC bond. According to the IR spectra, it is delocalized to form the 
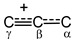
 moiety (Figure 6). The chemical shifts of its C atoms (230 (C_γ_), 150 (C_β_) and 215 ppm (C_α_)) are consistent with those of the 
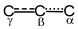
 part of vinyl cation **VII** (Figure 11). This means that electron density distributions—across the CC bonds of the two moieties, which differ in formal double and triple CC bonds—are of the same type, with a lower positive charge on the C_α_ atom than on atoms C_β_ and C_γ_. The observed participation of H atoms of the CH_3_ group in hyperconjugation with the partially empty 2p_z_ orbital of the C_α_ atom indicates its sp^2^ hybridization. That is, formally, the C_α_ atom must carry a considerable positive charge.

## 3. Methods and Materials

The carborane acids, H(CHB_11_Hal_11_) with Hal=F, Cl, were prepared as described previously [40,41]. They were purified by sublimation at 150–160 °C under pressure of 10^−5^ Torr on cold Si windows in a specially designed IR cell. The formed thin translucent layer yielded an intense IR spectrum. If this film is slightly wetted with a drop of 1,2-dichloroethane (DCE) and is allowed to dry quickly, the following reaction proceeds with the release of HCl:H{Hal_11_} + 2 C_2_H_4_Cl_2_ → (C_2_H_3_-Cl^+^-C_2_H_3_){Hal_11_^−^} + 3HCl.(1)

The amount of the formed HCl was determined as follows: the weighed portion of H {Cl_11_} was placed on the bottom of the IR cell and wetted with a drop of DCE. The quantity of released HCl was determined by measuring the intensity of its absorption at 2821 cm^−1^ with subsequent conversion to millimoles using a calibration curve. The determined molar ratio HCl/H {Cl_11_} is equal to 3 (Appendix A). Chemical analysis of the content of C and H in the salts of chloronium confirmed its chemical composition (Appendix A). When d_4_-DCE is used, reaction (1) proceeds with the formation of (C_2_D_3_-Cl^+^-C_2_D_3_) {Cl_11_^−^} and the release of both HCl and DCl (detected in IR spectra). Thus, reaction (1) proceeds via two steps:H{Cl_11_} + C_2_D_4_Cl_2_ → C_2_D_4_Cl{Cl_11_} + HCl, 
and
C_2_D_4_Cl{Cl_11_} + C_2_D_4_Cl_2_ → (C_2_D_3_-Cl^+^-C_2_D_3_){Cl_11_^−^} + 2 DCl.(2)

After completion of reaction (1), the product film formed on the surface of the Si windows has a yellow color due to the presence of impurities. These are easily removed by washing the film with cold dichloromethane (DCM), because they are readily soluble, whereas the main product dissolves slowly. The remaining film turns white and its IR spectrum confirms the removal of a small amount of impurities.

At ≥100 °C the chloronium cation decomposes into an unsaturated carbocation with the release of HCl, in accordance with Equation (3)
(C_2_H_3_-Cl^+^-C_2_H_3_){Cl_11_^−^} → C_4_H_5_^+^{Cl_11_^−^} + HCl,(3)
in a way similar to the mechanism of diethyl-chloronium decomposition into the *t*-Bu^+^ cation [32]. The thermal decomposition of deuterated divinyl-chloronium proceeds similarly, with the formation of the salt of the C_4_D_5_^+^ cation and the release of DCl.

The salts of ^13^C-substituted (C_2_H_3_)_2_Cl^+^ and C_4_H_5_^+^ cations were obtained via reactions (1)–(3) using ^13^C-dichloroethane (99 atom% ^13^C) from Sigma-Aldrich (Taufkirchen, Germany).

Solutions of carbocation salts in SO_2_ClF for NMR measurements were prepared under anaerobic conditions. A solid salt was placed at the bottom of an NMR tube. Through a capillary, gaseous SO_2_ClF was injected into it. At −15 °C, it was condensed into a liquid. The tube was sealed and shaken until a saturated solution was obtained. ^1^H and ^13^C NMR spectra of the solutions were recorded on a Bruker DRX 500 spectrometer (Heidelberg, Germany). Chemical shifts in the ^1^H NMR spectra were measured relative to an internal standard, the (C)H signal of the anion CHB_11_Cl_11_^−^, which is equal to 3.376 ppm in a CH_2_Cl_2_ solution referenced to TMS. Chemical shifts in the ^13^C NMR spectra were referenced to TMS as an external standard.

The preparation of samples for magic-angle spinning (MAS) NMR measurements is described in detail in the Appendix A.

The MAS NMR measurements were performed on a Bruker Avance-400 NMR spectrometer, Germany, (9.4 T) equipped with a broad-band double resonance MAS NMR probe (4 mm rotor diameter). The samples in the sealed glass ampoules were tightly inserted into MAS rotors and spun at 5–9 kHz. ^13^C MAS NMR spectra were recorded at ambient temperature at the resonance frequency of 100.63 MHz. The length of the 90° pulse was 6.5 µs, and 2000–10,000 scans were accumulated for a spectrum with a repetition delay of 10 s. To eliminate the effect of J-coupling between ^13^C and ^1^H nuclei, high-power proton decoupling was applied during the period of signal acquisition at the power level equivalent of a 90° ^1^H pulse of 5.0 µs length. ^1^H MAS NMR spectra were acquired at the frequency of 400.13 MHz by means of Hahn-echo pulse sequence (*π*/2–*τ*–*π*–*τ*–acquisition), where delay *τ* is equal to one rotor period, and the length of the *π*/2 excitation pulse is 5.0 μs. The ^1^H MAS NMR spectrum was accumulated after 16 scans with the repetition delay of 60 s, ensuring quantitative conditions. Chemical shifts in the ^13^C NMR spectra were referenced to TMS as an external standard and in the ^1^H NMR spectra relative to an internal standard: the (C)H signal of anion CHB_11_Cl_11_^−^, which is equal to 3.376 ppm in the CH_2_Cl_2_ solution referenced to TMS.

IR spectra were obtained on a Shimadzu IRAffinity-1S spectrometer (Japan) housed inside a glovebox in the 4000–400 cm^−1^ frequency range in transmittance and attenuated total reflectance mode (ATR). The spectra were manipulated using the GRAMMS/A1 (7.00) software from Thermo Scientific.

All the quantum chemical calculations were performed at the B3LYP/6-311G++(d,p) level plus the D3 dispersion correction energy term [42] with an ultrafine integration grid within the framework of the Gaussian’09 package [43].

## 4. Conclusions

Pure salts of the divinyl-chloronium cation, (C_2_H_3_)_2_Cl^+^, methylated propargyl (**VI**), and C_4_-allyl type **VII** carbocations with extremely stable and weakly basic carborane anions, CHB_11_Cl_11_^−^ and CHB_11_F_11_^−^, were obtained for the first time. They dissolved in DCM without interaction with the solvent but transformed into other carbocations. Therefore, they were studied in a solid phase by IR and ^1^H/^13^C MAS NMR spectroscopy. Only cation **VII** was stable in solutions of its salt in SO_2_ClF, which made it possible to obtain its detailed NMR spectra with hyperfine structure.

These cations are thermally stable at room and elevated temperatures (up to 100–110 °C for chloronium or ~200 °C for the methyl-propargyl carbocation). We reported the same earlier for vinyl-type carbocations [26,27,28], C_3_H_5_^+^ and C_4_H_7_^+^, thereby refuting the widespread belief that unsaturated (nonaromatic) carbocations are unstable at temperatures higher than −100 °C. The high stability of unsaturated carbocations is facilitated by their formal double and triple CC bonds, which contribute to the distribution of electron density over CC bonds and to the dispersion of a positive charge over the molecule.

The composition of the divinyl-chloronium cation is determined by the conditions of its preparation, and the structure is unambiguously proven by MAS NMR and IR spectroscopy. Its formal double C=C bonds have a bond status of less than one-and-half with low π-electron density between C-C atoms. At the same time, the electronic shielding of its C atoms is substantial, which is unexpected. The charge of the cation is concentrated mainly on the C_2_H_3_ groups. For this reason, it forms a contact ion pair with an anion through the CH group of one of the C_2_H_3_ groups, and this process considerably lowers its symmetry. All these features of the chloronium cation are indicative of its high polarizability.

In the C_4_H_6_Cl^+^ cation, the electron density distribution over CC bonds and the nature of the double C=C bond are very sensitive to the position of the Cl atom in the carbon chain, 
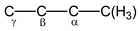
. When the Cl atom is attached to the C_α_/C_β_, atom, isomers 
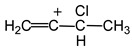
 and 
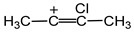
 arise, with well-defined and close-to-equivalent C=C double bonds. If the Cl atom is attached to the C_γ_ atom, then allyl-type isomer **VII** is formed with considerable alignment of the CC bonds in the 
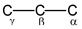
 moiety, with the charge dispersed over it. This phenomenon drives strong change in the CC stretch frequencies.

In propargyl carbocation **VI**, π-electron density of its formally triple C≡C bond is distributed to the neighboring C-C bond, thereby upgrading the neighboring bond to one-and-a-half-bond status and demoting itself to two-and-a-half-bond status. The asymmetry of the 
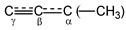
 moiety is evidenced by the higher frequency of the 
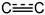
 stretch at 1563 cm^−1^ than that of the double C=C bond in vinyl cations (1490 cm^−1^) [26,27]. Despite the asymmetry, the positive charge is effectively distributed over both CC bonds: the involvement of the hydrogen atoms of the CH_3_ group in hyperconjugation with the 2p_z_ orbital of the C_α_ atom means that it has sp^2^ hybridization and a major positive charge is concentrated on it. Meanwhile, it follows from the MAS NMR spectra that a comparable charge is also localized on the C_γ_ atom.

Our important finding is that the charged formal triple C≡C bond in carbocation **VI** can accept an HCl molecule, giving rise to allyl cation **VII** with the Cl atom on a formal double C=C bond.

## Data Availability

Not applicable.

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
