# Peer review of "The Chloronium Cation [(C2H3)2Cl+] and Unsaturated C4-Carbocations with C=C and C≡C Bonds in Their Solid Salts and in Solutions: An H1/C13 NMR and Infrared Spectroscopic Study"

_ijms, 2022, doi:10.3390/ijms23169111_

Round 1

Reviewer 1 Report

In the manuscript, the authors’ statement is that they in first time obtained solid salts of the divinyl chloronium (C2H3)2Cl+ cation (I) and unsaturated C4H6Cl+ and C4H7+ carbocations with the highly stable CHB11Hal11- anion (Hal = F, Cl). In solid and solution respectively, the authors employed IR and 1H/13C MAS NMR spectra to characterize the conformational isomerism transformation with the charge and electron density re-distribution of chloronium cation and unsaturated C4-carbocations with C=C and C≡C bonds.

However, some comments should be addressed:

Comment 1:

The authors introduced that the solid salts of chloronium cation decompose into C4H5+ and the thermally stable carbocation is methyl propargyl. Moreover, the authors should show characteristic of temperature dependence of chloronium cation and the product from decomposing. Meanwhile, the chloronium cation in solution also should be investigated in different temperature.

Comment 2:

Either NMR or IR spectra, the authors just assigned several characteristic peaks. In fact, all assignments of all spectra should also be shown. For example in figure 2, the peak at 78.6 ppm of black line has been perturbed and even splitted in red line with or without high-power 1H decoupling. And the authors did not explain the reason.

Comment 3:

The authors wrote this manuscript irregularly. For example, an obvious error appeared in title like the H1/H13 NMR. The asterisks (*) were marked in several figures and tables without indicating what the mark means. In addition, this manuscript lacks supplementary information mentioned at the end of the text.

Author Response

Response to Reviewer 1

Review 1: Comments and Suggestions for Authors

In the manuscript, the authors’ statement is that they in first time obtained solid salts of the divinyl chloronium (C2H3)2Cl+ cation (I) and unsaturated C4H6Cl+ and C4H7+ carbocations with the highly stable CHB11Hal11- anion (Hal = F, Cl). In solid and solution respectively, the authors employed IR and 1H/13C MAS NMR spectra to characterize the conformational isomerism transformation with the charge and electron density re-distribution of chloronium cation and unsaturated C4-carbocations with C=C and C≡C bonds.

However, some comments should be addressed:

Comment 1:

The authors introduced that the solid salts of chloronium cation decompose into C4H5+ and the thermally stable carbocation is methyl propargyl. Moreover, the authors should show characteristic of temperature dependence of chloronium cation and the product from decomposing.

Response 1: We have studied the decomposition of chloronium cations in detail using the example of CH3-Cl+-CH3, CH2Cl-Cl+-CH2Cl, and CH3-Cl+-CH2Cl cations in solid salts and in their solutions in di-chloromethane (DCM) and published two large and complex articles [1] and [2] respectively. The thermal decomposition of the diethylchloronium salt (C2H5)2Cl+{Cl11-} at 140 C leads to the formation of t-Bu+{Cl11-} [3], while the thermal decomposition of the cyclic chloronium cation (CH2)4Cl+ - to the salt of the CH2=C+-CH2-CH3 cation [4]. The (CH2)4Cl+ cation, in solutions of its salts in DCM, spontaneously decomposes at room temperature to the final cations CH3- CH3-CHCl-C+=CH2 and CH3-C+=CCl-CH3, or CH3-C+=CH-CH3 [4]. In [3, 4], only the final cations formed during the decomposition of chloronium cations are discussed, since the process of decomposition of chloronium cations, as shown in [1, 2], is complex and requires deep study. Also in this article, we do not consider the process of decomposition of divinyl chloronium, as this is beyond the scope of this research. We study divinylchloronium only because we have obtained from it the product of interest to us, the methylated propargyl cation. The study of the process of its decomposition is not necessary for this work. This is a separate big job.  

[1] (a) Stoyanov, E. S. The salts of chloronium ions R–Cl+–R (R = CH3 or CH2Cl): formation, thermal stability, and interaction with chloromethanes. Phys. Chem. Chem. Phys., 2016, 18, 12896-12904. DOI: 10.1039/C6CP00946H; (b) Might also be helpful: Stoyanov, E. S. Chemical Properties of Dialkyl Halonium Ions (R2Hal+) and Their Neutral Analogues, Methyl Carboranes, CH3−(CHB11Hal11), Where Hal = F, Cl. J. Phys. Chem., A, 2017, 121, 15, 2918-2923. DOI: 10.1021/acs.jpca.7b01203).

[2] Stoyanov, E. S., Stoyanova I.V.  Cloronium Cations in Dichloromethane Solutions as Catalysts for the Conversion of CH2Cl2 to CHCl3/CCl4 and CH3Cl/CH4.  ChemistrySelect, 2018, 3, 12181 –12185. DOI: 10.1002/slct.201702738.

[3] Stoyanov E. S.; Stoyanova I.V.; Tham F.S.; Reed C. A. Dialkyl Chloronium Ions. J. Am. Chem. Soc., 2010, 132(12), 4062-4063.

[4] Stoyanov E. S., Bagryanskaya I. Yu., Stoyanova I.V. An IR-spectroscopic and X-ray-structural study of vinyl-type carbocations in their carborane salts. ACS Omega, 2022, DOI: 10.1021/acsomega.2c03025

Meanwhile, the chloronium cation in solution also should be investigated in different temperature.

Response: The divinylchloronium cation in solution of its salt in DCM rapidly decomposes at room temperature, and we were able to record its 1H NMR spectrum 2 min after preparation of the solution, simultaneously with a rather intense spectrum of decomposition products (Figure 4). The kinetics of its decomposition and the resulting products can be studied only at low temperatures by 1H NMR, and if possible, 13C NMR without IR spectroscopy. Our experience of studying of chloronium cations indicates that this is a complex work and does not correspond to the topic of this study. It can only be performed as an independent study.

Comment 2:

Either NMR or IR spectra, the authors just assigned several characteristic peaks. In fact, all assignments of all spectra should also be shown. For example in figure 2, the peak at 78.6 ppm of black line has been perturbed and even splitted in red line with or without high-power 1H decoupling. And the authors did not explain the reason.

Response 2. All signals of the NMR spectra are assigned: for the chloronium cation in Table 1, for cation VII in Table 4; for cation VI in Table 5. Thus, there is not a single unassigned signal in the NMR spectra of the studied carbocations. Regarding the peak at 78.6 ppm. In lines 96-98 it is written that two signals at 83.2 and 78.6 ppm belong to non-equivalent СαH groups and are split with a 1JCH constant of 304 Hz (Figure 2, red). “The other two signals obviously belong to carbon atoms of non-equivalent CβH2 groups. They must have a triplet structure. Nevertheless, due to the weaker influence of the positive charge on them, the spin–spin 13С-1Н constant decreases, thereby leading to signal broadening without multiplet resolution” (lines 98-101). For a signal of 43.9 ppm, the splitting is slightly pronounced. Therefore, we do not understand the reviewer's question that we did not explain the reason of the splitting.

About IR spectra. A complete interpretation is possible only for gas-phase IR spectra of naked cations in vacuum. The IR spectra of cations in the condensed phase always overlap with the strong spectrum of the anion, so it is fundamentally impossible to obtain the full spectrum of the cation. Only the most characteristic frequencies of CC, CH and CCl stretching vibrations, which are usually reliably detected, provide basic information about the cations under study. The remaining vibrations, as a rule, do not add significant information and therefore do not deserve to spend effort on their search and interpretation. That's why we work with the most useful CC, CH and CCl stretching vibrations (Tables 2, 3, 6). If we tried to interpret as many non-informative deformation vibrations as possible, then this would litter the article with information that is unnecessary for solving the tasks set. (The interpretation of these vibrations is hampered by the different asymmetries of the interaction of the cation with the anions of the nearest environment [26-28]).

-Comment 3:

The authors wrote this manuscript irregularly. For example, an obvious error appeared in title like the H1/H13 NMR.

Response. This is a very annoying typo. Corrected.

The asterisks (*) were marked in several figures and tables without indicating what the mark means.

Response. Figures 2, 4, 5, 6 contain (*) and in the captions under the figures their explanation is given. All other Figures (1,3,7,8,9,10,12,13) do not contain *, so the question about the Figures is not clear to us. The asterisks (*) are in Tables 1,2,5,6 and they are explained there (they are not in Tables 3,4). So, we didn't find any Tables containing not marked asterisks. We ask the Referee to indicate the numbers of Figures and Tables with not marked asterisks.

 In addition, this manuscript lacks supplementary information mentioned at the end of the text.

Response. In Supporting Information, the sentence “the IR spectrum of C4H5+{Cl11-} in CD2Cl2” was replaced with “the IR spectrum of the salt of isomer VII, isolated from SO2ClF solution.”

Reviewer 2 Report

The manuscript is described FTIR and NMR study of solid salts of the divinyl chloronium. The presented results contain novelty and originality. The manuscript could be published after the major corrections:

1) In the text and in Table 2 authors noticed bent vibration. What does it mean? Probably, it should be bending vibrations?

2) In my opinion, experimental section should be placed before Results and Discussion

3) In FTIR spectra authors should give the values on y-axis. What is absorption? Is this absorption coefficient in cm-1 or optical density (absorbance)?

4) In the experimental section authors described that FTIR spectra were measured in ATR regime. I have not found the ATR spectra in the Result. How were ATR spectra converted to Absorbance?

5) Some spectra (it is not clear how many?) were measured in transmittance mode. But thickness and geometry of the studied samples were not given.

6) In Figure 13 only one point does not correlate, why this is not systematic error? More discussion should be provided.

7) Methodology of ab initio calculation is absent. The results of the calculations are spreaded in the text and it is not clear.

Author Response

Response to Reviewer 2

Review 2: Open Review

The manuscript is described FTIR and NMR study of solid salts of the divinyl chloronium. The presented results contain novelty and originality. The manuscript could be published after the major corrections:

1) In the text and in Table 2 authors noticed bent vibration. What does it mean? Probably, it should be bending vibrations?

Response 1. Of course, it is bending vibrations. In Table 2, we corrected this, but in the text we found only one place (line 163) where a replacement needs to be made. 

2) In my opinion, experimental section should be placed before Results and Discussion

Response 2. We agree that the Experimental section can be placed before Results and Discussion. However, in the Rules for preparing articles for authors it is written: Research manuscripts should comprise sections: Introduction, Results, Discussion, Materials and Methods, Conclusions (optional). In all published articles in the IJMS that we have reviewed, Experimental section is placed after Results and Discussion. Therefore, we have tried to follow the Rules for preparing articles for authors. We do not know how strict these rules are and whether they can be broken. Therefore, I left it as it is for now, but in the cover letter to the editor I wrote: Is it possible to satisfy the reviewer's request and move the Experimental section before Results and Discussion? If he allows it, we will do it.

3) In FTIR spectra authors should give the values on y-axis. What is absorption? Is this absorption coefficient in cm-1 or optical density (Absorbance)?

Response 3. Of course, Absorbance. Corrections have been made. Figure 6 shows the ATR IR spectrum, for which it makes no sense to give numerical values on the y-axis. The values of the optical density along the y-axis in Figures 5 and 12 are not given, because the intensities of both IR spectra in the transmission of protium- and deuterium-samples are unified to the absorption intensity of their anion (this is stated in the caption under the figure: The intensities of both spectra are reduced to unit anion intensity).

4) In the experimental section authors described that FTIR spectra were measured in ATR regime. I have not found the ATR spectra in the Result. How were ATR spectra converted to Absorbance?

Response 4. ATR IR spectrum is given in the Figure 6. No ATR correction has been done for this spectrum because it is not compared with the spectrum in transmittance. Figures 5 and 12 shown IR spectra in transmission. Explanations have been added to the figure captions. IR spectrometer records ATR IR spectra in absorbance scale.

5) Some spectra (it is not clear how many?) were measured in transmittance mode. But thickness and geometry of the studied samples were not given.

Response 5. The thickness of the sample layer is given only when recording the spectra of liquid samples, and when recording solid samples, it is not given because it cannot be known. We did not record the spectra of liquid samples. The preparation of a thin layer of a solid sample for recording IR spectra in transmittance is described in sufficient detail in the Experimental part. We did not understand what is meant by the geometry of the studied samples.

6) In Figure 13 only one point does not correlate, why this is not systematic error? More discussion should be provided.

Response 6. The fact that all experimental and calculated frequencies for cation I correlate well with the exception of only one point, the frequency of C=C stretching, which deviates from the correlation line by 340 cm-1 (Figure 13) is an important result, meaning that quantum chemical calculations give a large error for the frequency of C=C bond (lines 436-437). The same deviation (360 cm-1) is observed for the cation I (line 440). All other studied vinyl cations have the similar features of the C=C stretching vibrations: ~160 cm-1 deviation for the isubutylene cation [26]; for cations H3C-CH=CH+ and (H2CCHCH2)+ they reach 362 and 286 cm−1 correspondingly [27]. The reasons for such low frequencies of C=C stretching are discussed in [28]. Such large deviations of the experimental C=C stretching frequencies from the calculated ones cannot be a systematic error in the experimental frequency determination, since it usually does not exceed 0.5 - 1 cm-1 for solid samples.  

We have made a small addition to the text starting from line 442.

7) Methodology of ab initio calculation is absent. The results of the calculations are spreaded in the text and it is not clear.

Response 7. The experimental part briefly presents the ab initio calculation methodology (Lines 594-596). In this work, calculations play a supporting role to determine the optimized structures of isomers in vacuum and their energeias (Schemes 1 and 3). We used a deeper application of quantum chemical calculations to vinyl cations in the works [26,27], where it was shown that the calculated frequencies of those groups of atoms where the positive charge is concentrated (mainly on C=C bonds) differ greatly from the experimental ones. Therefore, in the next work on vinyl cations [28], we did not use ab initio calculation referring to the results of [26,27]. In present work we confirmed that the calculated frequencies of the С≡С stretching vibration significantly exceeds the experimental one by 450 cm-1 (lines 477,478), i.e., their application for solving the problems of this work is very limited. In the future, it is necessary to conduct special studies by highly qualified quantum chemists to resolve this issue.  

Round 2

Reviewer 1 Report

No further comments

Reviewer 2 Report

The manuscript could be accepted